# RMSprop converges with proper hyper-parameter

**Naichen Shi** [*], **Dawei Li** [†], **Mingyi Hong**[‡] **Ruoyu Sun** [§]

## Abstract

Despite the existence of divergence examples, RMSprop remains one of the most popular algorithms in machine learning. Towards closing the gap between theory and practice, we prove that RMSprop converges with proper choice of hyper-parameters under certain conditions. More specifically, we prove that when the hyper-parameter $\beta_2$ is close enough to 1, RMSprop and its random shuffling version converge to a bounded region in general, and to critical points in the interpolation regime. It is worth mentioning that our results do not depend on "bounded gradient" assumption, which is often the key assumption utilized by existing theoretical work for Adam-type adaptive gradient method. Removing this assumption allows us to establish a phase transition from divergence to non-divergence for RMSprop.

Finally, based on our theory, we conjecture that in practice there is a critical threshold $\beta_2^*$, such that RMSprop generates reasonably good results only if $1 > \beta_2 \geq \beta_2^*$. We provide empirical evidence for such a phase transition in our numerical experiments.

## 1 Introduction

RMSprop (Tieleman & Hinton, 2012) remains one of the most popular algorithms for machine learning applications. As a non-momentum version of a more general algorithm Adam, RMSprop's good empirical performance has been well acknowledged by practitioners in generative adversarial networks (GANs) (Seward et al., 2018; Yazıcı et al., 2019; Karnewar & Wang, 2020; Jolicoeur-Martineau, 2019), reinforcement learning (Mnih et al., 2016), etc. In spite of its prevalence, however, Reddi et al. (2018) discovered that RMSprop (as well as the more general version Adam) can diverge even for simple convex functions. To fix the algorithm, the authors of Reddi et al. (2018) proposed a new variant called AMSGrad, which is guaranteed to converge under certain conditions.

Since then, it has been an active area of research to design provably convergent variants of RMSprop. These variants include AdaFom (Chen et al., 2019), Adabound (Luo et al., 2019), Nostalgic Adam (Huang et al., 2019), Yogi (Zaheer et al., 2018), and many more. Despite the variants, the vanilla RMSprop indeed works well in practice, and after proper hyper-parameter tuning, the non-convergence issue has *not* been commonly observed. Why is there a large gap between theory and practice? Is this because the real-world problems are likely to be "nice", or is it because the theoretical analysis of RMSprop does not match how it is used in practice?

With the above questions in mind, we revisited the counter-example of Reddi et al. (2018), and found an interesting phenomenon. One counter-example of Reddi et al. (2018) is the following:

$$f_{\text{t}}(x) = \begin{cases} Cx, & \text{for } t \bmod C = 1 \\ -x, & \text{otherwise} \end{cases} \tag{1}$$

where $x \in [-1, 1]$. They proved the divergence under the condition $\beta_2 \leq \min\{C^{-\frac{4}{C-2}}, 1 - \left(\frac{9}{2C}\right)^2\}$, where $\beta_2$ is the second order momentum coefficient in Algorithm 1 (the algorithm is presented later). For instance, when $C = 10$, then the algorithm diverges if $\beta_2 < 0.3$. Reddi et al. (2018) mentioned

---

[*]IOE, University of Michigan, naichens@umich.edu. Part of the work was done when Naichen Shi was working with Prof. Ruoyu Sun as an intern.

[†]ISE, University of Illinois at Urbana-Champaign. dawei2@illinois.edu

[‡]ECE, University of Minnesota - Twin Cities, mhong@umn.edu.

[§]University of Illinois at Urbana-Champaign. ruoyus@illinois.edu. Corresponding author: Ruoyu Sun.

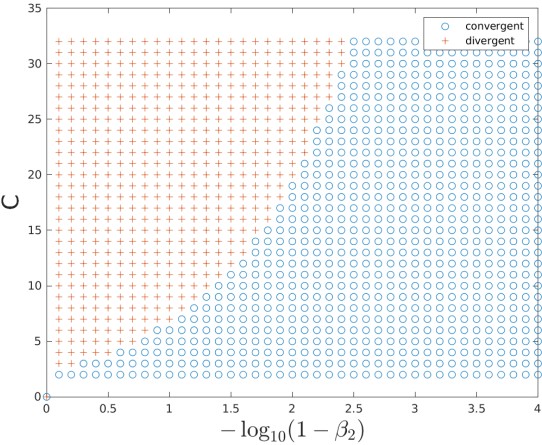

**Figure 1:** Phase diagram of the outcome of RMSprop on the counter example (1). Different marks represent different outcome: we label a data point as convergence if the distance between $x$ and $-1$ is smaller than 0.01 on average after 750000 iterations and as divergence otherwise. For each choice of $\beta_2$, there exists a counter example, but for each counter example in which Adam diverges, there exists a larger $\beta_2$ that can make Adam converge. We fix $\beta_1 = 0$. Step size is set as $\eta_t = \frac{1}{\sqrt{t}}$.

that "this explains why large $\beta_2$ is advisable while using Adam algorithm", but they did not analyze whether large $\beta_2$ leads to convergence in their example. We ran simulation for problem (1) with different $\beta_2$ and found there is always a threshold of $\beta_2$ above which RMSprop converges, see Figure 1. For instance, when $C = 10$, the transition point of $\beta_2$ is roughly 0.955: the algorithm converges if $\beta_2 > 0.956$ but diverges if $\beta_2 < 0.955$. In general, there is a curve of *phase transition* from divergence to convergence, and such a curve slopes upward, which means the transition point is closer to 1 if $C$ becomes larger. Based on this observation, we make the following conjecture:

> Conjecture: RMSprop converges if $\beta_2$ is large enough.

Before further discussion, we introduce the following assumption.

**Assumption 1.1.** $f(x) = \sum_{j=0}^{n-1} f_j(x)$, and

$$\sum_{j=0}^{n-1} \|\nabla f_j(x)\|_2^2 \leq D_1 \|\nabla f(x)\|_2^2 + D_0. \tag{2}$$

We divide optimization problems into 2 classes: *realizable* problems where $D_0 = 0$ and *non-realizable* problems where $D_0 > 0$. When $D_0 = 0$, the assumption (1.1) becomes $\sum_{j=0}^{n-1} \|\nabla f_j(x)\|_2^2 \leq D_1 \|\nabla f(x)\|_2^2$, which is called "strong growth condition" (SGC) (Vaswani et al., 2019). It requires the norm of the stochastic gradient to be proportional to the batch gradient norm. When $\|\nabla f(x)\| = 0$, under SGC we have $\|\nabla f_j(x)\| = 0$ for all $j$. For linear regression problems, SGC holds if the linear model can fit all data. More specifically, for the problem $\min_{\mathbf{x}} \|A\mathbf{x}\|^2 = \sum_{j=1}^{n} (a_j^T \mathbf{x})^2$ where $A$ is an $n$ by $n$ matrix and $a_j^T$ is the $j$-th row vector of $A$, SGC holds with $D_1 \leq \lambda_{\max} \left( \sum_{i=1}^{n} a_i a_i^T a_i a_i^T \right) / \lambda_{\min} (A^T A)$ (Raj & Bach, 2020). SGC can be viewed as a simple condition that models overparameterized neural networks capable of interpolating all data points (Vaswani et al., 2019). Therefore, in this work we use the terminology "realizable problems" to refer to the problems that satisfy SGC.

## 1.1 Main Contributions

In an attempt to resolve the conjecture, we delve into RMSprop's convergence issues and obtain a series of theoretical and empirical results. Our contributions are summarized below:

- We find that RMSprop's convergence is contingent on the choice of $\beta_2$. For general optimization problems, there are two types of hyper-parameters: *problem-dependent* hyper-parameters such as step size in GD, and *universal* constants such as momentum coefficient in heavy ball method [1]. Our result reveals that $\beta_2$ is closer to the first type.

- We prove that RMSprop converges to stationary point for realizable problems (interpolation regime), and to some bounded region for non-realizable problems. Combining with the divergence example of RMSprop, this indicates the existence of a phase transition from divergence to convergence dependent on $\beta_2$. Note that when we say "convergence", in a weak sense it means the sequence converges to a bounded region for non-realizable case; and in a strong sense it means the sequence converges to stationary points for realizable case.

- To our best knowledge, we are the first to prove the convergence of RMSprop and some of Adam *without* any form of assumption about the boundedness of the gradient norm. This is important for showing the transition: with added assumptions on bounded gradients, the gradients cannot diverge, while the counter-example shows that the gradient can.

## 2 PRELIMINARIES

We consider a finite-sum problem:

$$\min_{x \in \mathbb{R}^d} f(x) = \sum_{j=0}^{n-1} f_j(x). \tag{3}$$

In neural network training, $f_j$ usually represents the loss contributed by the $j$-th sample batch. We present randomly shuffled Adam in Algorithm 1. RMSProp is the special case of Adam with $\beta_1 = 0$. In this work, we mainly focus on RMSprop; nevertheless, we will present a result for a special case of Adam with small $\beta_1$.

---

**Algorithm 1** Randomly Shuffled Adam

---

Initialize $m_{1,-1} = \frac{1}{1-\beta_1} \nabla f(x_0)$ and $v_{1,-1} = \frac{1}{1-\beta_2} \max_j \{\nabla f_j(x_0) \circ \nabla f_j(x_0)\}$.
**for** $k = 1 \to \infty$ **do**
    Sample $\{\tau_{k,0}, \tau_{k,1}, \cdots, \tau_{k,n-1}\}$ as a random permutation of $\{0, 1, 2, \cdots, n-1\}$
    **for** $i = 0 \to n - 1$ **do**
        $m_{k,i} = \beta_1 m_{k,i-1} + (1 - \beta_1) \nabla f_{\tau_{k,i}}$
        $v_{k,i} = \beta_2 v_{k,i-1} + (1 - \beta_2) \nabla f_{\tau_{k,i}} \circ \nabla f_{\tau_{k,i}}$
        $x_{k,i+1} = x_{k,i} - \frac{\eta_{k*n}}{\sqrt{v_{k,i}}+\epsilon} \circ m_{l,k,i}$
    **end for**
    Break if certain stopping criterion is satisfied.
    $x_{k+1,0} = x_{k,n}, v_{k+1,-1} = v_{k,n-1}, m_{k+1,-1} = m_{k,n-1}$
**end for**
**return** x

---

In Algorithm 1, $x$ denotes the optimization variable, $m$ denotes the first order momentum and $v$ denotes the second order momentum. Specifically, we denote $x_{k,i}, m_{k,i}, v_{k,i} \in \mathbb{R}^d$ as the value of $x, m, v$ at the $k$-th outer loop and $i$-th inner loop, respectively. We denote $\nabla f_j$ as the gradient of $f_j$ and let $\circ$ be the component-wise multiplication. The division of two vectors is component-wise as well. Moreover, we denote $\eta_t$ as the step-size and $\beta_1, \beta_2$ as the hyper-parameters in the algorithm. When $n = 1$, we obtain full batch Adam.

We replaced the bias correction step in (Kingma & Ba, 2015) with a special initialization on $m_{1,-1}$ and $v_{1,-1}$. This initialization can also correct the bias, but has cleaner results. Since the effect of initialization or bias correction becomes more and more negligible as the training progresses, RMSprop with zero initialization or our initialization will have the same asymptotic behavior. We put our results for the original version of RMSprop in the appendix.

---

[1] Rigorously speaking, for the best convergence rate, the momentum coefficient should also be problem-dependent; but just for achieving convergence, it can be problem independent.

As for hyper-parameters, we choose $\eta_t = \frac{\eta_1}{\sqrt{t}}$ and fix $\beta_2$ to be a constant that is independent of the iteration count. We allow $\epsilon$ to be an arbitrary non-negative constant; in particular, our result holds even for $\epsilon = 0$. The constant $\epsilon$ is added in practice for numerical stability, and $\epsilon$ is typically chosen to be $10^{-6}$ or even $10^{-8}$. It is much smaller than $\sqrt{v_{k,i}}$ (which is roughly the size of gradient norm).

## 2.1 RELATED WORK

As discussed earlier, one line of research focuses on variants of RMSprop and Adam that can be proved to converge. These works usually modify the update rule of $v_t$. For instance, AMSGrad (Reddi et al., 2018), AdaFom (Chen et al., 2019) explicitly make $v_t$ non-decreasing. Nostalgic Adam (Huang et al., 2019) and the algorithms analyzed in Zou et al. (2019) and Chen et al. (2019) use iteration-dependent $\beta_{2t}$ (and/or $\beta_{1t}$) to let $v_t$ weigh more on past gradients. Some works add new modifications into RMSprop and Adam; for instance, Zhou et al. (2019) mitigate the bias in update direction by using a different estimate of $v_t$, Dozat (2016) combine Adam with Nesterov momentum, and Liu et al. (2020a) employ a warm-up technique.

Besides modifying the algorithm, a few attempts have been made to address the non-convergence issues of the original versions, but they often rely on extra assumptions. A number of works (Zaheer et al., 2018; De et al., 2019; Défossez et al., 2020) prove the convergence of Adam under these additional assumptions. One representative work along this line, Défossez et al. (2020), establishes a clean convergence result and also provides some insights on the momentum mechanisms by improving the dependence of the iteration complexity on $1 - \beta_1$. However, these works assume $\epsilon$ to be relatively large compared to $\sqrt{v_{k,i}}$. The issue is that such a choice essentially transforms RMSprop back to SGD since the effective step size is primarily controlled by $\epsilon$, in lieu of $\sqrt{v_{k,i}}$. This is in contrary to the spirit of RMSprop, which is to use adaptive step size to accelerate convergence. A few other works do not need the assumption of $\epsilon$, but they have other assumptions. De et al. (2018) analyze deterministic and stochastic RMSprop, but they utilize a rather unrealistic assumption that the sign of all noisy gradients are the same, i.e., $\text{sign}(\nabla f_p(x)) = \text{sign}(\nabla f_q(x))$ for all $p, q$. Chen et al. (2019) describe a few quantities based on the iterates, and prove that if they grow in a certain speed as the iterates go, the algorithm converges. The drawback is that the condition cannot be checked *a priori*. Besides the assumptions mentioned above, all the aforementioned works require the gradient to be bounded.

In general, removing boundedness assumptions (of any kind, including bounded gradient, bounded iterates, etc.) is not necessarily easy. Thus, such results are appreciated even for basic SGD. For instance, Bertsekas & Tsitsiklis (2000) presents a nice discussion of various results on inexact GD without involving conventional bounded assumptions, and claims "bounded-assumption-free" as one of the main contributions of their work. Very recently, we notice another work (Liu et al., 2020b) which removes the bounded gradient assumption for SGDM (SGD with momentum) and obtains satisfactory rates. Nevertheless, we are not aware of an existing result on RMSprop that does not require bounded gradient assumption. We will explain later why removing this bounded gradient assumption is particularly important for our paper.

## 3 THE *raison d'être* FOR $\beta_2$

Figure 1 clearly demonstrates the important role of $\beta_2$ in the convergence of RMSprop. Specifically, a sufficiently large $\beta_2$ is critical for RMSprop's convergence. Indeed, some recent works (Reddi et al., 2018; Zhou et al., 2019) have also made similar arguments, but they focus on understanding one part of the phenomenon, that is, small $\beta_2$ leads to divergence. Our goal in this work is to complete the other part of the story by showing that, sufficiently large $\beta_2$ guarantees convergence. The formal result will be provided in Sec. 4.

To understand the function of $\beta_2$, we first discuss why RMSprop diverges. It is known that the stochastic noise due to mini-batch will distort the gradient direction, leading to possible divergence, but in standard SGD, the distortion in multiple iterations is eliminated since the stochastic gradient is an unbiased estimate of the gradient. For RMSprop, at a given iteration the scaling constant $1/\sqrt{v}$ in the update direction may cause larger gradient distortion than the standard SGD. The distortion can be so significant that the average updating direction falls outside the dual cone of the true gradient. To illustrate this, consider the extreme case that $\beta_2 = 0$ and $\epsilon = 0$ (i.e., signSGD) and the special

example (1). When applying signSGD to solve (1), in each epoch which consists of $C$ iterations, one iteration will move $x$ left followed by $C - 1$ iterations that move $x$ right. Since all step sizes are the same in one epoch, the accumulated effect of one epoch makes $x$ move in the ascending direction, instead of the descending direction.

Then why does large $\beta_2$ help? Intuitively, a large $\beta_2$ can control the distortion on update directions. In the extreme case that $\beta_2 = 1$ and $\epsilon = 0$, RMSprop reduces to SGD where the distortion of multiple iterations can be mitigated, leading to convergence. We suspect that $\beta_2$ does not need to be exactly 1, and a large $\beta_2$ is enough to control the distortion. Our experiment in Figure 1 confirms that, at least for the counter-example of Reddi et al. (2018), there is an interval $\beta_2 \in [c, 1]$ such that RMSprop converges.

What was initially not clear is whether the counter-example of Reddi et al. (2018) is a very special case or the convergence of large-$\beta_2$-RMSprop holds for all problems. We found the real situation is somewhat more tricky. For *non-realizable* problems, we discovered an example for which RMSprop cannot converge to the minimum for a wide range of $\beta_2 < 1$, but unlike the small-$\beta_2$-case the iterates converge to a small ball around the minimum. This motivates us to distinguish three convergent-situations: divergence, convergence to a small region, convergence to critical points. What we can prove for the general problem is (see Theorem 4.3): for small $\beta_2$, RMSprop can diverge; for large $\beta_2$, RMSprop must converge to a small region whose size depends on $\beta_2$.

Then why do we observe the convergence to a single point in the experiment for (1)? We suspect this is because the problem (1) is realizable, and conjecture that the property of "convergence to critical points" holds for all realizable problems. We indeed prove this conjecture (see Corollary 4.1): large-$\beta_2$-RMSprop converges to critical points if the problem satisfies SGC.

We summarize our findings about the convergence properties of RMSprop in Table 1.

| Setting | $\beta_2$ close to 1 | $\beta_2$ close to 0 |
|---|---|---|
| non-realizable | converge to small region (Thm. 4.3) | diverge |
| realizable | converge to critical points (Coro. 4.1) | diverge |

**Table 1:** Outcome of shuffled RMSprop under different hyperparameter settings. On unconstrained problems, if the gradient norm can decrease to zero after infinite iterations, we classify the result as "convergence to critical points". If the gradient norm keeps increasing to infinity during training, we classify the result as "divergence". We classify a result as "converge to small region" if the gradient norm falls below a certain level but not close to 0.

Note that our results do not conflict with Theorem 3 in Reddi et al. (2018) which claims that "for any constant $\beta_1$ and $\beta_2$ there exists a divergent example" since here we choose $\beta_2$ to be problem-dependent, just like one chooses a step size $< 2/L$ for GD where $L$ is a problem dependent parameter. Another remark is that though $\beta_2$ could be close to 1, RMSprop still retains the ability to adapt $v$ to gradient square norm as long as $\beta_2 < 1$, because new gradient signals are added for each iteration and the impact of previous signals decays exponentially. It is the adaptive ability that distinguishes RMSprop from SGD. Proving the theoretical advantage of RMSprop over SGD (i.e., choosing $\beta_2 < 1$ is better than $\beta_2 = 1$) is a very intriguing question; in general, the theoretical advantage of adaptive gradient methods (including RMSprop and AdaGrad) over SGD is a long standing question. in this work, we focus on the fundamental problem of convergence, instead of the more challenging question of justifying the advantage of RMSprop.

## 4 CONVERGENCE RESULTS

In this section, we present the formal theoretical results. We start from the results of full batch RMSprop/Adam, and then present the results for the stochastic versions. Note that random shuffling is not a key factor, and the proof works for other settings.

## 4.1 FULL-BATCH VERSION

We first consider the full-batch version of RMSprop. The following theorem shows that if we use all samples to evaluate the gradient, RMSProp with diminishing stepsize converges to critical points regardless of the choice of $\beta_2$. Here we consider one popular step size schedule that $\eta_t = \frac{\eta_1}{\sqrt{t}}$.

**Theorem 4.1.** *(convergence of full-batch RMSprop) For problem (3) with $n = 1$, assume that $f$ is gradient Lipschitz continuous with constant $L$ and lower bounded by $f^*$. Then, for full-batch RMSprop (Alg. 1 with $\beta_1 = 0, \epsilon = 0$) with diminishing step size $\eta_t = \frac{\eta_1}{\sqrt{t}}$ and any $\beta_2 \in (0, 1)$, we have:*

$$\min_{t \in (1,T]} \|\nabla f_t\|_1 \leq \mathcal{O}\left(\frac{\log T}{\sqrt{T}}\right)$$

*where $T > 0$ is the total iteration number.*

De et al. (2019) also proves the convergence of full batch RMSprop, but they require the gradient norm to be upper bounded; in contrast, we do not need this assumption, and only require lower-boundedness and $L$-smoothness of $f$.

Our result suggests that the convergence property of batch-RMSprop is similar to signSGD, an algorithm that only uses the sign of gradient to calculate its descent direction (Bernstein et al., 2018): in the full-batch setting, signSGD (which can be called sign GD) has also been proved to converge without bounded gradient assumption.

Below, we also derive an analogous result for full-batch Adam with only one additional constraint $\beta_1 < \sqrt{\beta_2} < 1$, which is often satisfied in practice:

**Theorem 4.2.** *(convergence of full-batch Adam) For optimization problem (3) with $n = 1$, assume that $f$ is gradient Lipschitz continuous with constant $L$ and lower bounded by $f^*$. Then, for full-batch Adam with diminishing step size $\eta_t = \frac{\eta_1}{\sqrt{t}}$ and any $\beta_1 < \sqrt{\beta_2} < 1$, we have:*

$$\min_{t \in (1,T]} \|\nabla f_t\|_1 \leq \mathcal{O}\left(\frac{\log T}{\sqrt{T}}\right).$$

## 4.2 STOCHASTIC VERSIONS

As mentioned earlier, our simulation shows that RMSprop may not converge to critical points for non-realizable problems (an example is provided in the appendix). Nevertheless, we can still show randomly shuffled large-$\beta_2$-RMSprop converges to a bounded region:

**Theorem 4.3.** *(large-$\beta_2$ RMSprop converge to a region) For problem (3), assume $f$ is lower-bounded by $f^*$ and all $\nabla f_j$ is $L$-Lipschitz continuous. Furthermore, assume (2) holds, and $\beta_2$ satisfies*

$$T_2(\beta_2) \triangleq \sqrt{\frac{10dn}{\beta_2^n}} dn D_1 \left((1 - \beta_2)\frac{\left(\frac{4n^2}{\beta_2^n} - 1\right)}{2} + \left(\frac{1}{\sqrt{\beta_2^n}} - 1\right)\right) \leq \frac{\sqrt{2} - 1}{2\sqrt{2}}, \qquad (4)$$

*Then, for randomly shuffled RMSprop with $\eta_t = \frac{\eta_1}{\sqrt{t}}$, we have*

$$\min_{t \in (1,T]} \min\{\|\nabla f_{nt}\|_1, \|\nabla f_{nt}\|_2^2 \sqrt{\frac{D_1 d}{D_0}}\} \leq \mathcal{O}\left(\frac{\log T}{\sqrt{T}}\right) + \mathcal{O}\left(Q_{3,3}\sqrt{D_0}\right), \ \forall T \geq 4.$$

*Here $Q_{3,3} > 0$ is a $\beta_2$-dependent constant that goes to zero in the limit as $\beta_2 \to 1$.*

**Remark 1.** This result and the result in Reddi et al. (2018) together distinguish large-$\beta_2$-RMSprop and small-$\beta_2$-RMSprop: the former converges to a bounded region, while the latter can diverge. Note that there is a gap between the lower bound of $\beta_2$ and the upper bound of $\beta_2$ in the counter-example. We do not try to provide tight bounds on the threshold of $\beta_2$, as our main goal is to show a qualitative difference between large-$\beta_2$-RMSprop and small-$\beta_2$-RMSprop.

**Remark 2**:

Condition (4) in Theorem 4.3 implies that $1 - \beta_2 \leq \mathcal{O}\left(n^{-3.5}\right)$. In the appendix we introduce three problem-dependent parameters $\rho_1 \in [1, \sqrt{n}]$, $\rho_2 \in [0, n]$, and $\rho_3 \in [1, \sqrt{n}]$ in equations(14), (15)

and (16), and improve the sufficient condition (4) to $1 - \beta_2 \geq \mathcal{O}\left(1/\left(n\rho_1\rho_2\rho_3\right)\right)$. For the worst case, the bound is $\mathcal{O}\left(n^{-3.5}\right)$, just like condition (4) in Theorem 4.3. In actual training process, $\rho_1, \rho_2$, and $\rho_3$ may not reach their upper bounds, thus the threshold of $\beta_2$ can be lower in practice (see Appendix A.5 for some empirical estimate of $\rho_i$'s). The dependence on the number of batches $n$ suggests that as $n$ increases, the required hyper-parameter $\beta_2$ should be larger. This is understandable since more minibatches means larger noise in the stochastic gradient, and thus larger $\beta_2$ is required. There is a gap between our theoretical bound of $\beta_2$ and the empirical transition point, and it is an interesting future question to close this gap.

**Remark 3.** We point out three possible algorithm behaviors: divergence to infinity (or divergence for short), convergence to a bounded region (or non-divergence for short) and convergence to critical points. We distinguish the three cases, making it easier to explain the qualitative difference of small-$\beta_2$ and large-$\beta_2$ regime. For non-realizable cases, the phase transition is from divergence to non-divergence. Therefore, it is important to discard the bounded-gradient assumption: this assumption eliminates the possibility of divergence of gradients a priori. To be clear, there are actually two sub-cases of non-divergence: iterates can stay in a bounded but huge region (bad case), or iterates stay in a bounded region dependent on some parameters (good case). Indeed, the "convergence" of constant-stepsize SGD is in the sense of "converging to a region with size proportional to the noise variance". Our result of "converging to bounded region" is also meaningful as the size of the region goes to zero as the noise variance goes to 0 or $D_0$ goes to 0 (realizable case).

Note that "divergence" can be also interpreted as "not converging to critical points" which is the notion used in Reddi et al. (2018), instead of "diverging to infinity". We use the latter concept of "diverging to infinity" for the term "divergence", because "not converging to critical points" can include the good case of converging to a small region around critical points (like constant-stepsize SGD). In the example of Reddi et al. (2018), a constrained problem is considered (bound constraint [-1,1]), thus divergence to infinity cannot happen. We add an example where the iterates and the gradients can diverge to infinity for small $\beta_2$; see Appendix A.2.

As a corollary of Theorem 4.3, if the optimization problem satisfies SGC (i.e. $D_0 = 0$), RMSprop converges to critical points.

**Corollary 4.1.** *Suppose the assumptions of Theorem 4.3 holds. Further, assume* (2) *holds with* $D_0 = 0$, *i.e.,* $\sum_{j=0}^{n-1} \|\nabla f_j(x)\|_2^2 \leq D_1 \|\nabla f(x)\|_2^2$ *for all* $x$*. we have:*

$$\min_{t \in (1,T]} \|\nabla f_{nt}\|_1 \leq \mathcal{O}\left(\frac{\log T}{\sqrt{T}}\right), \ \forall \, T \geq 4.$$

With the above corollary, the numerical result in Figure 1 should not be surprising: problem (1) satisfies the strong growth condition, and thus there is always a range of $\beta_2$ inside which RMSprop converges. We just need to tune $\beta_2$ larger.

We can prove similar convergence results for Adam with small $\beta_1$ and large $\beta_2$.

**Theorem 4.4.** *For optimization problem (3), assume that* $f$ *is lower-bounded by* $f^*$ *and* $f_j$ *is gradient Lipschitz continuous with constant $L$ for all $j$. Furthermore, assume that $f_j$ satisfies (2) for all $x$. Then, for randomly shuffled Adam with diminishing step size* $\eta_t = \frac{\eta_1}{\sqrt{t}}$ *and* $\beta_1, \beta_2$ *satisfying* $T_1(\beta_1, \beta_2) + T_2(\beta_2) < 1 - \frac{1}{\sqrt{2}}$, *we have* $\min_{t \in [1,T]} \|\nabla f_{nt}\|_1 \leq \mathcal{O}\left(\frac{\log T}{\sqrt{T}}\right) + \mathcal{O}\left(Q_{3,5}\sqrt{D_0}\right) \ \forall \, T \geq 4$, *where $Q_{3,5}$ is a constant that approaches 0 in the limit $T_1 + T_2 \to 0$, $T_2$ is defined in (4), and $T_1$ is defined as* $T_1(\beta_1, \beta_2) = \sqrt{\frac{5dn}{\beta_2^n}} dn^2 D_1 \frac{\beta_1}{\beta_2^n} \left(\frac{1-\beta_1}{1-\beta_1^n} + 1\right)$.

Remark: This result shows that controlling $\beta_2$ and $\beta_1$ together can ensure convergence of Adam. We conjecture that the same convergence can be proved for a large range of $\beta_1$, but we are not able to prove that for now (which is why we focus on RMSprop in this work) and leave it to future work.

## 5 EXPERIMENTS

We conduct experiments of image classification and GAN on MNIST and CIFAR-10 to support our theoretical findings. The details of GAN experiments are in the appendix, and in this section we focus on image classification results.

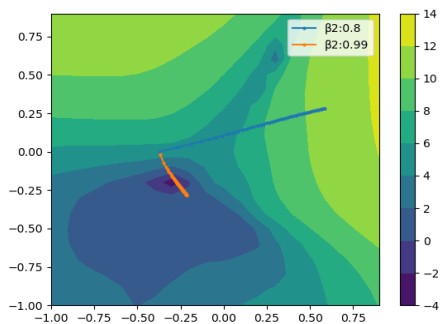

**Figure 2:** Trajectories of RMSprop with large or small $\beta_2$ on the loss surface. Two trajectories start from the same initialization and are trained on the same dataset MNIST, but have very different behavior. We calculate the hyper-plane spanned by the starting point of two trajectories and their respective ending points, and then project two trajectories onto this plane. The large-$\beta_2$-trajectory converges to minima, while the small-$\beta_2$-trajectory diverges. The difference between these two trajectories suggests a phase transition for $\beta_2$ in $(0.8, 0.99)$.

We visualize the optimization trajectory when training on MNIST for small $\beta_2 = 0.8$ and large $\beta_2 = 0.99$ in Figure 2. We observe different behaviors: while the trajectory of $\beta_2 = 0.8$ moves away from the bottom of the basin, for larger $\beta_2$ the trajectory stays in the level set and has decreasing loss values.

In the CIFAR experiments, we use ResNet-18. We choose $\beta_2 = 0.8, 0.9, 0.95, 0.99$ respectively. With different batch sizes $8, 16, 32$, we run each algorithm for 100 epochs without explicit regularization. Table 2 shows two phenomena: first, for fixed batch size, there is a transition point of $\beta_2$ above which the accuracy suddenly jumps; second, the transition point is closer to 1 as the batch size decreases. More specifically, for batch size 8, the transition point lies in $[0.95, 0.99]$: the average training accuracy is 44.53% for $\beta_2 = 0.95$, but jumps to 99.74% for for $\beta_2 = 0.99$. For batch size 16, the transition point lies in $[0.9, 0.95]$: the average training accuracy is 67.27% for $\beta_2 = 0.9$, but jumps to 96.38% for $\beta_2 = 0.95$. For batch size 16, the transition point lies in $[0.8, 0.9]$. As batch size increases from 8 to 16 and then 32, then transition point decreases from 0.99 to 0.95 and then to 0.9.

These two phenomena are consistent with our theory. The first phenomenon can be explained by Theorem 4.3 and Corollary 4.1 which state that for large enough $\beta_2$, RMSprop converges. The second phenomenon can be explained by Theorem 4.3 as well: as explained in Remark 2, the required $\beta_2$ decreases as the number of mini-batches $n$ decreases, i.e., the batch size increases.

**Table 2:** Performance of Adam with different $\beta_2$ with ResNet-18 on CIFAR-10 (100 epochs)

| batch size | measure | $\beta_2$=0.8 | $\beta_2$=0.9 | $\beta_2$=0.95 | $\beta_2$=0.99 | SGD |
|---|---|---|---|---|---|---|
| 8 | train acc. | 10.00±0.00 | 10.00±0.00 | 44.53±32.09 | 99.74±0.06 | 100.00±0.00 |
|  | test acc. | 10.00±0.00 | 10.00±0.00 | 42.02±29.59 | 70.23±0.26 | 70.37±0.45 |
| 16 | train acc. | 28.70±32.39 | 67.27± 8.98 | 96.38±1.35 | 99.75±0.05 | 99.98±0.02 |
|  | test acc. | 27.64±30.55 | 62.71±7.71 | 70.11±0.90 | 70.43±0.15 | 69.45±0.38 |
| 32 | train acc. | 66.93±3.07 | 96.72±1.36 | 99.17±0.42 | 99.80±0.14 | 81.50±1.57 |
|  | test acc. | 62.99±2.13 | 70.05±1.40 | 71.92±0.50 | 71.34±0.60 | 68.92±1.12 |

Next, we demonstrate that the convergence speed of SGD is much slower than Adam under the same experiment setting as Table 2. We compare the average training and test accuracy at the 10-th epoch. As Table 3 shows, the accuracy of Adam is much higher than SGD at the 10-th epoch.

All codes generating experimental results are available on the Github repository `https://github.com/soundsinteresting/RMSprop`

**Table 3:** Training and test accuracy at the 10-th epoch

| batch size | measure | $\beta_2$=0.99 | SGD |
|---|---|---|---|
| 16 | train acc. | 95.41±0.81 | 65.89±1.28 |
| | test acc. | 70.02±0.17 | 62.62±1.26 |
| 32 | train acc. | 97.92±0.23 | 57.87±0.70 |
| | test acc. | 70.44±0.19 | 56.18±0.86 |

## 6 Conclusion

In this work, we study the convergence behavior of RMSprop by taking a closer look at the hyper-parameters. Specifically, for realizable problems, we provide a data-dependent threshold of $\beta_2$ above which we prove the convergence of randomly shuffled RMSprop and small $\beta_1$ Adam without bounded gradient assumption. We also show that RMSprop converge into a bounded region under non-realizable settings. These findings reveal that there is a critical threshold of $\beta_2$ regarding the convergence behavior of RMSprop, and the phase transition is supported by the numerical experiments. Our results provide basic guidelines for tuning hyper-parameters in practice.

## 7 Acknowledgement

M. Hong is supported by NSF grant CMMI-1727757. Ruichen Li from Peking University helped check some proof of Theorem 4.3. We thank all anonymous reviewers for their feedback. We also want to thank Eduard Gorbunov and Juntang Zhuang for pointing out some mistakes on openreview in the earlier versions.

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
