# OpenReview forum: "RMSprop converges with proper hyper-parameter"
_ICLR.cc/2021/Conference — ICLR 2021 Spotlight_

### Official Review · AnonReviewer4 · 2020-10-26
**The paper studies one of the most popular algorithms in machine learning: RMSprop. The proposed problem is practical. Overall, I vote for accepting the paper.**

**Rating:** 6
**Confidence:** 3

**Review:**

Summary:

The paper studies one of the most popular algorithms in machine learning: RMSprop. More specifically, it investigates the relation between the hyper-parameters and the convergence of the algorithm. By proving the convergence without using "bounded gradient" assumption, the authors establish a phase transition from divergence to non-divergence for RMSProp.

Pros:

1. The paper concerns one of the most important algorithms in machine learning. In my opinion, the problem is practical and of interest in machine learning community.

2. The results of the paper provide explicit conditions for the hyper-parameters of RMSprop/Adam that ensure the convergence of the algorithms. These results provide basic guidelines for tuning hyper-parameters of the algorithms in practice.

Cons:

Apart from the strong points, I still have some concerns about the clarity of the paper. I hope the authors can address my concerns to improve the quality of the paper.

1. The parameter $\beta_2$ is the most important subject of the paper. Until algorithm 1, the paper discusses $\beta_2$ without defining it clearly. It would be more clear if $\beta_2$ is mentioned from the beginning of the paper that it comes form algorithm 1.

2. The authors divide the problems into 2 sub-classes to investigate: realizable and non-realizable, which are not clearly defined. It would be better if the authors can define these 2 sub-classes more formally.

3. The experiments supporting the theoretical results are comprehensible. However, I would suggest the authors provide a figure with x-axis to be epochs and y-axis to be accuracy so that the readers can have better idea upon how SGD and RMSProp behave during training.

---

> ### Author Response · Authors · 2020-11-25
> **Thank you for your comments!**
>
> Thank you for your comments! Below we provide the responses to specific comments.
>
> 1. Definition of $\beta_2$:
>
> That’s a good suggestion! We now modified the structure as follows: we added an explanation that β2 is from Algorithm 1 in the beginning of the introduction, as suggested by the reviewer.
>
> 2. Formal definition of realizable and non-realizable problems
>
> Thanks for the suggestion. We moved the formal definitions from the section on the main result to the end of the introduction section. As stated there, we say a problem is realizable if it satisfies condition (3) with $D_0=0$, and non-realizable if it satisfies condition (3) with $D_0>0$.
>
> 3. Figure illustrating performance of algorithms during training
>
> Thank you for the suggestion. We followed the suggestion and now included a figure in Appendix A.1; see Figure A1 in the modified version of the paper.

---

### Official Review · AnonReviewer1 · 2020-10-28
**Interesting analysis of the role of the beta2 parameter in RMSProp convergence behaviour**

**Rating:** 8
**Confidence:** 3

**Review:**

The paper starts off from the recent realization that there exists divergent examples for any set of hyperparameters for algorithms in the Adam family, such as RMSProp. It sets out to study the effect of the beta2 parameter on convergence for a fixed specific problem. The analysis shows that there exists a beta2 < 1 that leads to convergence for realizable problems, and to convergence to a bounded region of interest for non-realizable problems, without requiring a bounded-gradient assumption. Experiments confirm this new theory.

Overall, the paper is well-written, clear and easy to read. One of its strongest points is how well the analysis and the relevance of the results is motived. For instance, the importance of removing the assumption on the bounds on the gradient because it effectively removes one of the convergence/divergence regimes is well executed.
There is also significant efforts on providing clear simplified examples from rather complex theorems, which is very appreciated (e.g. Corollary 4.1).
Further, there is a real effort to contrast the results with the previous work, and to explain how it complements them, resolving clearly what initially appears as direct contradictions.

The results are relevant, both from the point of view of the theory, where it adds to a body of work explaining how and why the Adam family of algorithm performs well on modern machine learning taskloads, and from the point of view of the practitioner, outlining what hyperparameter tuning is necessary to achieve convergence. They are also original, in the sense that they provide novel insights, while removing problematic assumptions that permeate most of the related work.

A couple of things could be improved:
- as pointed out in the paper, if beta2 = 1, the algorithm degenerates to SGD. While there is a remark explaining why as long as beta2 < 1 the two algorithms differ, it would be informative to compare the convergence regimes with high beta2 to SGD directly, to validate that there exists a set of hyperparameters that not only provide convergence, but improved convergence properties compared to SGD (otherwise the results are a lot less relevant), as well as give an order of magnitude of what value is typically necessary for beta2.
- condition (4) in theorem 4.3 is quite difficult to apprehend, with a slightly worrying beta2^n term. More exegesis would be beneficial for reader comprehension.

Overall, this is a nice, well-written and relevant paper that clears the bar for publication in its current version.

---

> ### Author Response · Authors · 2020-11-25
> **Thank you for the detailed and constructive comments!**
>
> Thank you for the detailed and constructive comments! Below we provide the responses to specific comments.
>
> 1. RMSProp and SGD
>
> Thanks for the suggestion. Our convergence rate in high $\beta_2$ regime is $O\left(\frac{\log T}{\sqrt{T}}\right)$, and under the nonconvex setting, the best proved rate of SGD is also $O\left(\frac{\log T}{\sqrt{T}}\right)$ (Saeed Ghadimi and Guanghui Lan. Stochastic first-and zeroth-order methods for nonconvex stochastic programming). Thus, the two rates are theoretically comparable.
>
> We agree that proving “adaptive gradient methods have improved convergence properties compared to SGD” is a very interesting problem.  In our paper and other previous works such as Mnih et al., 2016, Seward et al., 2020; Yazıcı et al., 2019,  the advantage of RMSprop over SGD is mainly demonstrated through experiments. The intuition of the advantage of adaptive gradient methods is that by using adaptive learning rate, we can better utilize the local structure of the problem and thus achieve faster convergence, which is similar to the idea of Quasi-Newton method. Nevertheless, even for convex problems, the theoretical advantage of the Quasi-Newton method was not well understood (a possible benefit is asymptotic superlinear convergence, but such a benefit is not easy to prove for stochastic versions). Thus it seems that providing a strong theoretical justification for the advantage of Adam-type methods is challenging and requires much more work. We hope to explore this problem in the future.
>
> 2. Exigesis on condition (4):
>
> Thanks for the comment. The estimate of $\beta_2$ by condition (4) in theorem 4.3 is $1-\beta_2\le O\left(\frac{1}{n^{3.5}}\right)$. We did not try to optimize this bound before as we focus on proving the existence of such a threshold. We agree that explaining the bound will help readers understand RMSProp better, so we add some discussions on $\beta_2$ below and in the revised version. The bound indicates that the transition point is roughly $\beta_2 = 1 - 1/poly(n)$.  In our experiments on Resnet18 with batchsize 8 and n $\approx$ 5000, the transition point of $\beta_2$ is between $0.95$ and $0.99$; for n $\approx 2500$ (batch size 16), the transition point lies in [0.9,0.95];
> for $n\approx 1200$, the transition point lies in [0.8,0.9]. This indicates that the transition point is decreasing over $n$, which qualitatively matches our bound of $\beta_2$. In this experiment, the practical size of the $\beta_2$-threshold
> seems to be somewhere between $1 - O(1/\sqrt{n})$ and $1 - O(1/n)$. Note that we do not have extensively studied the order of the transition point, so this is a preliminary empirical estimate. If the empirical estimate is reasonable, then there is a gap between our theoretical bound and this empirical estimate.
>
> In an effort to reduce the gap, we revisited our proof, and find that our proof only requires $1-\beta_2\le O\left(\frac{1}{n\rho_1\rho_2\rho_3}\right)$, where $\rho_1$, $\rho_2$, and $\rho_3$ are three quantities that characterize the distribution of the gradients; see equations (14)-(16), equation (37) and the remark after it in the appendix of the revised version. The upper-bounds of $\rho_1$, $\rho_2$, and $\rho_3$ are $\sqrt{n}$, $n$, and $n$ respectively, thus in the worst case the bound is $1-\beta_2\le O\left(\frac{1}{n^{3.5}}\right)$. This is why we obtain this bound in the original condition (4). The benefit of introducing the quantities $\rho_1$, $\rho_2$,  and $\rho_3$ is that the threshold of $\beta_2$ may be smaller than the worst-case estimate if $\rho_1$, $\rho_2$,  and $\rho_3$ are small in practice. For instance, if $\rho_i = O(1)$, then  $1 - \beta_2 \approx O(1/n). $ We checked the size of $\rho_i$'s in a MNIST experiment, and find that $\rho_1 \rho_2 \rho_3$ roughly lies in $[ 1, n ]$, in which case the bound of $\beta_2$ is somewhere between $1- O\left(\frac{1}{n^{1}}\right)$ and $1- O\left(\frac{1}{n^{2}}\right)$. This is closer to the emprical transition point we observe. Anyhow, it remains a question to provide a more precise estimate of $\beta_2$.
>
> We add a remark (Remark 2 after Theorem 4.3) in the revised version to explain the size of $\beta_2$ and the stronger bound based on $\rho_i$'s; we also provide a preliminary empirical estimate of $\rho_i's$ in Appendix A.5.

---

### Official Review · AnonReviewer3 · 2020-10-28
**Very interesting work on classical adaptive methods**

**Rating:** 8
**Confidence:** 3

**Review:**

This work revisits a famous counterexample on the convergence of Adam (originally presented in Reddi 2018). The authors show that, if the EMA parameter beta2 in RMSprop and Adam is chosen high enough, then both methods converge to a bounded region in the stochastic setting. In addition, the authors provide some results for the full-batch case. Crucially, and differently from many other papers on the topic, the gradients are not assumed to be bounded and the beta2 hyperparameter is not chosen to increase to 1.

The paper is well written and the logic of it is convincing. I like the introduction and Figure 1 (this nicely illustrates the relevance of this paper). It is also very well organized. Unfortunately, I did not have the time I wish I had to dig into the proofs (just had a quick check), but the methodology of the authors and the results are convincing.

This is overall a very nice paper, with clean and easy to read results, that clarifies an important point: it is misleading to claim that “Adam does not converge” (which was pointed out in Reddi 2018 to introduce AMSgrad). I have heard this (wrong) claim many times in the optimization community – hence I think this paper deserves attention (therefore my clear accept). This work truly does merge the gap between theory and practice in non-convex stochastic optimization.

Just a few suggestions: I think the authors should cite and discuss the results in Defossez et al. 2019 (On the convergence of Adam and Adagrad). Also, I think Figure 1 deserves better quality. It's done in matlab so in the xlabel command you can put 'interpreter','latex' and 'fontsize',20. Finally, I spotted 1 typo: in Remark2 “cases of non-divergence cases”.

---

> ### Author Response · Authors · 2020-11-25
> **Thank you for your supportive and encouraging comments.**
>
> Thank you for your supportive and encouraging comments! Below we provide the responses to specific suggestions.
>
> 1. The work Defossez et al. 2019 On the convergence of Adam and Adagrad：
>
> Thank you for the suggestion. That is a representative paper and we now added the discussion of it in the related work section: “The work (Defossez et al. 2019) establishes a clean convergence result and also provides some insights on the momentum mechanisms by improving the dependence of $1-\beta_1$ in the convergence rate. However the work still assumes bounded gradient and non-zero $\epsilon$.”
>
> 2. x-label of figure 1：
>
> Thanks a lot for your suggestion! We have replotted Figure 1 according to your suggestion.
>
> 3. Typo：
>
> Thanks for pointing it out. We now changed it to “cases of non-divergence”.

---

### Comment · ~Juntang_Zhuang1 · 2021-01-16
**Questions regarding toy examples**

Thanks for the nice paper. Very good theoretical analysis, however I'm somehow confused by the toy examples.

(1) Figure 1 corresponds to the example $f_(x) = Cx\  \textit{if t mod C}==1$, but when I check the code, you are actually using $t mod C <=1$ in the reddieexample.m function, and for each period of C the function observes two gradients of C rather than 1. The actual region of convergence is much smaller than it shows in Figure 1. Please clarify if I misunderstand it.

Furthermore, when I change the code to t mode C ==1, and I found for a large region, say C >=15, I found it not converge even beta2 is set as 1-10^{-4}. Is this a numerical issue, or does this imply that there exists some C, such that no matter how close beta2 to 1, RMSProp just won't converge?

(2) For Fig.A4 in appendix A.4, should it correspond to the non-realizable toy-example in A.4? In text it's said "FigA5". Furthermore, I'm confused by the text that "SGD converges, AMSgrad converges, RMSprop fail to converge". If you look at the y-axis, which is log(1-x^*), SGD result is around 0.5, which is much worse than RMSProp (-0.8) and AMSGrad (-1.5), though there's a trend that RMSProp increases the distance, SGD and AMSGrad decreases the distance, but can we safely draw the conclusion that "RMSProp fail to converge, SGD converges" here?

Thanks a lot in advance

---

> ### Author Response · Authors · 2021-01-19
> **Thanks for your comment**
>
> Hi Juntang:
>
> Thank you for your comments! Below are some preliminary responses:
>
> (1) Thanks for pointing that out. We modified the code accordingly in the repository and replotted figure 1 (which will be updated soon). The actual range of convergence is indeed smaller than the what was shown in the initial version of figure 1. But still, for every C, there is a range of $\beta_2$ that makes RMSprop converge. Specifically, for the case you mentioned, C=30 and $\beta_2=0.999$, if you run RMSprop for 200k iterations using diminishing stepsize $\frac{1}{\sqrt{t}}$, you will find that $|x-x^\star|$ is converging (in the last few iterations, this gap is below 0.01 and decreasing).  Nevertheless, if only run 20k iterations, then the error is still quite large, so it looks like it is not converging.
>
> (2) It is FigA.4 that corresponds to the toy-example in A.4; thanks for pointing out the typo. We will correct it in the revised version.
>
>   For the concern whether the experiments are enough to show RMSprop does not converge and SGD converges: thank you for pointing this out. First, SGD provably converge with diminishing stepsize. Nevertheless, since "convergence" does not mean fast convergence, it may take many iterations for $|x-x^\star|$ to reach small error, which is the case for our example. Second, RMSprop also moves slowly for our example, and we suspect it may take more than $10^{12}$ iterations for RMSprop to achieve error larger than SGD. We did not run that many iterations but speculate according to the trend based on the results of $10^7$ iterations. The slow training of the two methods is the reason why SGD does not achieve smaller error than RMSprop.
>
>   Previously, we thought the trend of the two methods (RMSProp goes up, and SGD goes down) is enough. After your comment, we realized that we could have made the experiments more convincing.
>
>   The slow training is largely because we chose $a =10000$. There are two ways to alleviate the concern: i) run $10^{12}$ iterations for the original example; ii) still run $10^7$ iterations, but for a simpler problem. We choose the second way: we modify the problem from $a = 10000$ to $a=100$. Then the experiments show what we expect: SGD achieves error smaller than RMSProp after $4\times 10^6$ iterations and is further going down, while RMSProp error is still going up after $10^7$ iterations.  The related code is in our repository now; you can also simply replace $a=10^4$ by $a=10^2$. The modified experiments and changes will be updated into our revised version.
>
>   Again, thank you for your comments which help improve the paper. Feel free to let us know any questions.

---

> > ### Comment · ~Juntang_Zhuang1 · 2021-01-24
> > **Thanks for clarification**
> >
> > Thanks for clarification. Looking forward to some more discussion regarding the second question.

---

### Comment · ~Eduard_Gorbunov1 · 2021-05-02
**Question about Theorem 4.3**

Dear authors,



Thank you for the paper! Removing bounded gradient assumption from the analysis of RMSprop is a really good contribution. However, one place in your paper is not clear to me. In Theorem 4.3, you write that $Q_{3,3} \to 0$ when $\beta_2 \to 1$, meaning that one can make the last term in your upper bound arbitrary small via increasing $\beta_2$. From the proof of Theorem 4.3 I derived that $Q_{3,3} \sim C_5$ (proportional to $C_5$) where the second term in $C_5$ is proportional to $\sqrt{\frac{T_2}{1-\beta_2}}$. Next, the first term in $T_2$ is proportional to $\frac{1-\beta_2}{\beta_2^n}\cdot \left(\frac{4n^2}{\beta_2^n} - 1\right)$. Therefore, we get that $\sqrt{\frac{T_2}{1-\beta_2}} \sim \sqrt{\frac{1}{\beta_2^n}\cdot \left(\frac{4n^2}{\beta_2^n} - 1\right)}$ that goes to some positive constant when $\beta_2 \to 1$. Other terms in $Q_{3,3}$ are non-negative, meaning that $Q_{3,3} \to C > 0$ when $\beta_2 \to 1$.



Have I misunderstood something? Could you please clarify this part of your paper? Thank you in advance!



Best regards,
Eduard Gorbunov

---

> ### Author Response · Authors · 2021-05-04
> **Thanks for your comment**
>
> Dear Eduard:
>   Thanks for pointing this issue out. In the current form, $Q_{3,3}$ indeed converges to some nonzero constant in the limit $\beta_2\to1$.
>   The dependence of $Q_{3,3}$ on $\beta_2$ is a minor claim in theorem 4.3, and does not affect other parts of the paper.
>   However, we are still improving the proof so that the additional error term will reduce to 0 when $\beta_2\to 1$, as is suggested by some experimental observations. We will post the refined version very soon.

---

> > ### Comment · ~Eduard_Gorbunov1 · 2021-05-05
> > **Thanks for the reply**
> >
> > Dear authors,
> >
> > Thank you for the reply. I am looking forward to the new version.
> >
> > Best regards,
> > Eduard Gorbunov

---

### Comment · ~Kyunghun_Nam1 · 2023-10-13
**Question about Appendix A.4**

Dear authors,

I hope this message finds you well.
Thank you for your nice research and paper.
However, I have some queries about A.4 in the appendix

Can you tell me how you set the value of $\beta_2$ in this graph?

** Please, double-check the last line of the same subsection.
"Moreover, when $\beta_2$ becomes closer to zero, the distance is smaller."

---

> ### Author Response · Authors · 2023-10-14
> **Thank you for your question**
>
> Dear Kyunghun Nam:
>
>   Thank you for your interest in our paper! We set $\beta_2$ sufficiently large to be in the non-divergence regime. More specifically, we set $\beta_2=0.999$ in this experiment. You can check our Github repository for the Matlab code:
>
> https://github.com/soundsinteresting/RMSprop/blob/master/nonrealizable.m
>
>   Thank you for pointing out the typo in the last sentence. It should be: Moreover, when $\beta_2$ becomes closer to $1$, the distance is smaller.

---

> > ### Comment · ~Kyunghun_Nam1 · 2023-10-15
> > **Additional question about notations and some expressions**
> >
> > I would like to ask an additional question regarding the notation in the paper.
> > The notation in Appendix C says that $x_{l, t}$ is the $l$-th component of $x$ at epoch $t$ in the full-batch case.
> > Then, by Lemma C.1, $|x_{i, t+1} - x_{i, t}|$ denotes the difference between the $i$th component of $x$ at epoch $t+1$ and epoch $t$ if $x$ is a full batch.
> > But what does this mean if it's a stochastic version?
> > * Is it the $i$-th iteration of the $t+1$th epoch or the $i$-th component of the $t+1$th epoch vector $x$?
> >
> > In Appendix F, I see the following notation defined and I'm not sure I understand it, can you talk me through it again?
> > * $b_{l. k} = \arg \max_{i \in \{0, 1, \cdots, n-1 \}} |g_{l, k, 0, i}|$ and $g_{l, k}^b = g_{l, k, 0, b_{l, k}}$.
> >
> > * In the proof of Lemma F.1.(Proof of theorem 4.3), I don't quite understand the meaning of "the last 4 lines" and how this is derived. Additionally, what does $\eta_{nk}$ mean and why does $\frac{\eta_{1} n}{\sqrt{nk}}$ disappear from $\frac{1}{\sqrt{1 - \beta_2}}$ in the last line?
> >
> > * And, in the same subsection, You assume that  $|\partial_{l} f(x_{k, 0})| \ge \frac{\eta_1 L \sqrt{dn} n^2}{\sqrt{k}\sqrt{1 - \beta_2}} \left( \right)$.
> > However, in the proof of Case 1 ( $M < k-1$ ), in the sentence "Since we assumed $|\partial_l f(x_{k, 0})| \ge \frac{\eta_1 L n \sqrt{dn}}{\sqrt{k} \sqrt{1 - \beta_2}} \left( \right)$, the largest coordinate of the gradient is also lower bounded by $|g_{l, k}^b| \ge \frac{\eta_1 L \sqrt{dn}}{\sqrt{k}\sqrt{1 - \beta_2}} \left( \right)$."
> > The three lower bounds that should be the same are all different. Please let me know which one is correct, if this is a typo, or if I missed something.
> >
> > And, in the same part (Case 1), I'm not sure how the following inequality came about.
> >
> > * $|g_{l, k}^b| \le \frac{\Delta_1 2 (M+2) \sqrt{n}}{\sqrt{k+1}} < \frac{\Delta_1 4M \sqrt{n}}{\sqrt{k}}$.
> >
> > Applying the upper bound of Lemma C.3 as follows, I could only get the following inequality.
> >
> > * $|g_{l, k}^b| \le \sum_{j=1}^{M+1} \sqrt{n} \Delta_{k-j} \le \sqrt{n} \Delta_1 2 \frac{M+1}{\sqrt{k-1}}$.
> >
> >
> > Sorry for the many questions.
> > Thank you in advance.

---

> > > ### Author Response · Authors · 2025-02-21
> > > **Thank you for your questions**
> > >
> > > Dear Kyunghun,
> > >
> > > Thank you very much for your questions. Below, we will answer each question separately.
> > >
> > > Theorem 4.3 is an almost sure convergence result. We analyze the worst case convergence behavior of a random shuffling algorithm, and prove the convergence in the worst case.
> > >
> > > In lemma C.1 and C.2, $x_{i,t}$ means the $i$-th coordinate of variable $x\in \mathbb{R}^d$ at iteration $t$. An epoch consists of multiple ($n$) iterations. Here, $t$ is an index of the iteration number.
> > >
> > > We defined $g_{l,k,i,j}=\frac{\partial}{\partial x_l} f_j(x)|{x=x_{i,k}}$ at the beginning of Appendix C. For the random shuffling algorithm, at the beginning of $k$-th epoch, we calculate the $g_{l,k,0,i}$ for all batch index $i$ and choose the largest index $i$ to be $b_{l,k}$.
> > >
> > > We defined our stepsize $\eta_t=\frac{1}{\sqrt{t}}$. Therefore, $\eta_{nk}=\frac{1}{\sqrt{nk}}$.
> > >
> > > We know $|g^b_{lk}|\ge \frac{1}{n}|\partial_lf(x_{k,0})|$. Therefore, under the assumption $|\partial_lf(x_{k,0})|\ge \frac{\eta_1L\sqrt{dn}n^2}{\sqrt{k}\sqrt{1-\beta_2}}$, all inequalities you mentioned will hold.
> > >
> > > You are correct about the upper bound of $|g^b_{lk}|$. The derivation should be $|g^b_{lk}|\le \sqrt{n}\Delta_12\frac{M+1}{\sqrt{k-1}}\le \frac{\Delta_14M\sqrt{n}}{\sqrt{k}}$. This is because $\sqrt{k-1}\ge\frac{\sqrt{k}}{\sqrt{2}}$ for $k\ge 1$ and $M+1\le \sqrt{2}M$ for $M\ge 3$.

---

> > ### Comment · ~Kyunghun_Nam1 · 2023-10-15
> > **Additional question about theorem 4.3**
> >
> > Thank you for your kind and quick response.
> >
> > Let me ask you one additional question.
> > For the stochastic version of RMSprop (Theorem 4.3) , there is no Expectation ($\mathbb{E}$) on the LHS, and I was wondering if you could explain why this is possible.
> > Because since $x_t$ is a stochastic process, there are 3 forms of convergence we can have, and the one you presented doesn't seem to be one of them.
> >
> > 1. Expectation form (e.g., $\min_{t} \mathbb{E} \nabla f(x_t)$)
> > 2. With a probability form (e.g. with a probability at least $1 - \delta$)
> > -> In this case, the RHS must contain $\delta$.
> > 3. Almost sure form
> >
> > Additionally, what is the meaning of $\nabla f_{nt}$? $nt$-th iterations or $nt$-th epochs?
> >
> > If there's something I'm missing, please let me know. Thank you

---

### Author Response · Authors · 2025-02-21
**Corrections of typo**

We correct two typos in our proof.

- On page 20 of the appendix, the two summation inequalities should be
$$
\sum_{t=t_{init}}^T\frac{1}{t}\le \log\frac{T}{t_{init}-1},
$$
and
$$
\sum_{t=t_{init}}^T\frac{1}{\sqrt{t}}\ge 2\left(\sqrt{T+1}-\sqrt{t_{init}}\right)
$$

Therefore, by setting $t_{init}=2$, we can derive the convergence result at the end of Appendix D as,

$$
\min_{t\in[2,T]} \lVert\nabla f_t\rVert_1\le \frac{1}{\sqrt{T+1}-\sqrt{2}}\left( Q_{1,1}+Q_{2,1}\log T\right)
$$

- The results on page 22, page 37, and page 42 should be adjusted accordingly.

---

### Decision · Program_Chairs · 2021-01-07
**Final Decision**

**Decision:**

Accept (Spotlight)

**Comment:**

The paper shows convergence results for RMSprop in certain regimes. The reviews are uniformly positive about this paper and I recommend acceptance.